# Global seroprevalence of Zika virus in asymptomatic individuals: A systematic review

**Paola Mariela Saba Villarroel**[1,2], **Rodolphe Hamel**[1,2,3], **Nuttamonpat Gumpangseth**[1,2], **Sakda Yainoy**[1], **Phanit Koomhin**[4], **Dorothée Missé**[3], **Sineewanlaya Wichit**[1,2]*

**1** Department of Clinical Microbiology and Applied Technology, Faculty of Medical Technology, Mahidol University, Nakhon Pathom, Thailand, **2** Viral Vector Joint unit and Joint Laboratory, Mahidol University, Nakhon Pathom, Thailand, **3** MIVEGEC, Univ. Montpellier, CNRS, IRD, Montpellier, France, **4** Center of Excellence in Innovation on Essential Oil, Walailak University, Nakhonsithammarat, Thailand

* sineewanlaya.wic@mahidol.ac.th

**Data Availability Statement:** The data supporting the findings of the study are available within the manuscript and in the supplemental information.

## Abstract

### Background

Zika virus (ZIKV) has spread to five of the six World Health Organization (WHO) regions. Given the substantial number of asymptomatic infections and clinical presentations resembling those of other arboviruses, estimating the true burden of ZIKV infections is both challenging and essential. Therefore, we conducted a systematic review and meta-analysis of seroprevalence studies of ZIKV IgG in asymptomatic population to estimate its global impact and distribution.

### Methodology/Principal findings

We conducted extensive searches and compiled a collection of articles published from Jan/01/2000, to Jul/31/2023, from Embase, Pubmed, SciELO, and Scopus databases. The random effects model was used to pool prevalences, reported with their 95% confidence interval (CI), a tool to assess the risk of study bias in prevalence studies, and the $I^2$ method for heterogeneity (PROSPERO registration No. CRD42023442227). Eighty-four studies from 49 countries/territories, with a diversity of study designs and serological tests were included. The global seroprevalence of ZIKV was 21.0% (95%CI 16.1%-26.4%). Evidence of IgG antibodies was identified in all WHO regions, except for Europe. Seroprevalence correlated with the epidemics in the Americas (39.9%, 95%CI:30.0–49.9), and in some Western Pacific countries (15.6%, 95%CI:8.2–24.9), as well as with recent and past circulation in Southeast Asia (22.8%, 95%CI:16.5–29.7), particularly in Thailand. Additionally, sustained low circulation was observed in Africa (8.4%, 95%CI:4.8–12.9), except for Gabon (43.7%), and Burkina Faso (22.8%). Although no autochthonous transmission was identified in the Eastern Mediterranean, a seroprevalence of 16.0% was recorded.

### Conclusions/Significance

The study highlights the high heterogeneity and gaps in the distribution of seroprevalence. The implementation of standardized protocols and the development of tests with high

**Funding:** This work was financially supported by the international postdoctoral fellowship 2022 provided by Mahidol University (P.M.S.V) and the National Research Council of Thailand (NRCT): NRCT5-RGJ63012-125, Grant No. RGNS 64-172 by Office of the Permanent Secretary, Ministry of Higher Education, Science, Research and Innovation (MHESI) (N.G.). The funders had no role in the study design, data collection, analysis or decision to publish, nor in the preparation of the manuscript.

**Competing interests:** The authors have declared that no competing interests exist.

specificity are essential for ensuring a valid comparison between studies. Equally crucial are vector surveillance and control methods to reduce the risk of emerging and re-emerging ZIKV outbreaks, whether caused by *Ae. aegypti* or *Ae. albopictus* or by the Asian or African ZIKV.

## Author summary

Zika virus (ZIKV) remains an important public health concern. Estimating the true burden of the disease is a major challenge, often underestimated due to the substantial number of asymptomatic infections. Consequently, seroprevalence studies are valuable for determining the geographic extent of the virus, measuring levels of human immunity, and assessing potential infection-related risks. Thus, we conducted a systematic review and meta-analysis of the literature addressing the seroprevalence of ZIKV in asymptomatic individuals worldwide. The overall seroprevalence of ZIKV IgG antibodies was 21.0%, calculated based on 84 studies published between January 2000 and July 2023. Antibodies have been found in all WHO regions, but none in Europe, with rates ranging from 8.4% in Africa to 39.9% in America. Further standardized surveillance studies are needed to understand immunity per region and over time, evaluate vector dynamics, and assess the risk of future outbreaks.

## Introduction

Zika virus (ZIKV) is an arthropod-borne virus classified within the *Flaviviridae* family. It is mainly transmitted to humans through the bite of infected *Aedes aegypti* and *Aedes albopictus* mosquitoes. The first human ZIKV infections dates back to the 1950s. Subsequent serological studies revealed sporadic infections in African and Asian countries and territories over the following five decades. However, a transformative event unfolded after May 2007 when ZIKV emerged as an important human pathogen responsible for substantial human outbreaks in the Pacific Islands, followed by an explosive and rapid dissemination across the Americas during 2015–2016. Phylogenetic analysis has identified two major ZIKV lineages: the African lineage (East and West African sublineages), and the Asian lineage (Asian and American sublineages), with the latter being responsible for the recent outbreaks [1–3].

ZIKV continues to pose a significant global threat, characterized by its persistent geographic expansion. According to the most recent data available, a total of 89 countries and territories, among five of the six WHO regions have documented autochthonous ZIKV transmission. The Eastern Mediterranean Region stands as the only region without documented autochthonous ZIKV transmission [4].

Approximately 80% of ZIKV infections typically manifest as asymptomatic, and when symptoms do occur, they are usually self-limited and short-lasting. However, during recent ZIKV disease outbreaks, complications including Guillain-Barré syndrome (GBS) and congenital Zika syndrome (CZS) have been documented. Additionally, other modes of transmission have been described, aggravated by the absence of specific treatments or effective vaccines [5].

The current disease surveillance system heavily relies on data collected from health centers and laboratories, primarily focusing on symptomatic cases. While this approach facilitates the early detection of outbreaks, it is not designed to provide an accurate estimation of the true

disease burden. In light of these limitations, seroprevalence studies assume a pivotal role in estimating population immunity, providing insights into geographic impact and distribution. Moreover, they contribute significantly to the evaluation and guidance of disease prevention and control strategies. Nevertheless, conducting seroprevalence studies poses significant challenges, mainly due to the extensive serological cross-reactivity observed between flaviviruses, notably with dengue virus (DENV). Consequently, additional tests to improve specificity are necessary, such as seroneutralization methods. However, these methods are characterized by being time-consuming, costly, labor-intensive, and are often not readily available [1,6].

The objective of our study was to perform a systematic review and meta-analysis of published studies on the seroprevalence of ZIKV IgG or IgM/IgG antibodies in asymptomatic populations between January 2000 and July 2023. Our study aimed to provide an update of a previous systematic review [6] conducted in a limited number of studies (n = 12) for a better understanding of the global distribution, prevalence rates, risk factors, and changes over time of ZIKV.

## Methods

This systematic review and meta-analysis was conducted in accordance with the PRISMA guidelines (S1 Table). This study is registered on the PROSPERO international prospective register of systematic reviews platform, under registration number CRD42023442227.

### Search strategy

We conducted a comprehensive and exhaustive search for articles published between January 01, 2000, and July 31, 2023 from Embase, Pubmed, SciELO, and Scopus databases. To perform this search, we utilized the following Medical Subject Headings (MeSH) terms: "Zika virus", "arboviruses", "seroepidemiologic studies", "serologic", "prevalence". These search terms were combined using the Boolean operators, AND and OR. Our search was limited to studies involving humans and those published in English, French, Spanish or Portuguese languages. Additionally, we included relevant papers obtained from the references of the original articles. To ensure comprehensiveness, we also conducted a free search using the Google search engine.

### Inclusion and exclusion criteria

We included cohort and cross-sectional studies of more than 88 participants in the sample size, except in cases where articles covered different time periods. Our inclusion criteria encompassed studies involving individuals from the general population, as well as specific population from all over the world. We considered studies that utilized any serological techniques to detect past ZIKV infection (IgG or IgM/IgG). The exclusion criteria comprised articles that assessed the same set of samples (the most complete version was included), reviews, studies involving febrile patients, studies without differentiation between flaviviruses, suspected or confirmed infections, those not covering the study period, research involving travelers, migrants or non-human subjects, as well as case control studies.

### Literature selection and data extraction

After removing duplicates, the titles and abstracts were screened by a review team member (PMSV). Full texts that met the potential inclusion criteria were evaluated by the same member (PMSV) and an independent reviewer (RH) to assess their eligibility based on the inclusion and exclusion criteria. Any disagreements were resolved through discussion with a third member (SW). Irrelevant articles were excluded, and the reasons for their exclusion were documented.

Data were extracted into an Excel spreadsheet, including the author's name, year of publication, sampling period, sample size, study design (cohort, cross-sectional), study location, setting (urban, rural, semi-urban), study population (general and specific population), age groups (adults [$\geq$13 years old] and children [<13 years old]), proportion of female participants, diagnostic method, ZIKV strain (Asian or African), number of positive cases, and data interpretation. Furthermore, any available risk factors for positivity were extracted. In cases involving multinational studies or studies conducted in different periods, the information was separated by country and/or year. Data were categorized according to World Health Organization (WHO) regions, which include Africa, the Americas, the Eastern Mediterranean, Europe, Southeast Asia, and the Western Pacific. Additionally, they were grouped into different sampling periods, including the entire period, as well as specific time frames (Jan 2000-Apr 2015 and May 2015-Jul 2023).

## Study quality assessment

The bias risk in the included studies was assessed using a modified version of the tool designed by Hoy *et al*., 2012, which includes a 10-item checklist [7]. Each item was assigned a score of "0" for "no" or "unclear" and "1" for "yes". A total score of 7–10 was considered "low risk", 4–6 "moderate risk", and 0–3 "high risk" of bias. Two authors independently evaluated the risk bias (PMSV and RH). Any disagreements were resolved through discussion and consultation with a third author (SW).

## Statistical analysis

Continuous variables were presented as means and standard deviation (SD), while categorical variables were expressed as numbers and percentages. The seroprevalences and their corresponding 95% confidence intervals (CIs) were calculated using the abstracted data.

In the meta-analysis, the $I^2$ method was employed to assess heterogeneity. An $I^2$ value of 25.0%, 50.0%, and 75.0% indicated low, moderate, high heterogeneity, respectively. The data were converted using Freeman-Tukey transformation to satisfy the normal distribution assumption. A random-effects model was chosen for studies with moderate to high heterogeneity; otherwise, a fixed-effects model was used. Furthermore, Egger's test was used to evaluate the potential presence of publication bias.

The proportion of ZIKV seropositivity in each study was combined to derive a pooled seroprevalence of ZIKV worldwide. The data were further categorized and analyzed according to WHO regions and based on sampling years (before and after the first ZIKV case detected in Sao Paulo, Brazil in May 2015). Moreover, subgroup analyses were performed based on the study population, age groups, settings, diagnostic methods, and ZIKV strains.

The statistical analyses were performed using SPSS 13.0 software (SPSS, Inc., Chicago, IL). The χ2 test was used for group comparisons, considering statistical significance when the p-value was <0.05. To generate forest plots, calculate CIs, and evaluate heterogeneity, JBI Sumari and MedCalc software's were used. Additionally, maps were created using Mapchart.net to provide a better visualization of seroprevalence estimates per country and administrative divisions, as well as the number of studies conducted in each country.

## Results

### Literature search

The process of selecting studies reporting ZIKV seroprevalences is illustrated in Fig 1. Initially, a database search yielded a total of 1682 records. After removing duplicates, 798 articles

underwent evaluation based on their titles and abstracts, resulting in the identification of 104 articles that were eligible for full-text review. Among these studies, those that did not meet the inclusion criteria were subsequently excluded, leading to a final inclusion of 84 articles with 113 data points when divided by year and country.

## General study characteristics and quality assessment

Seroprevalence studies were conducted in 49 countries and territories, with a total of 63,864 individuals involved. These studies comprised 18 countries in Africa (including Burkina Faso [8], Cape Verde [9], Cameroon [10], Democratic Republic of the Congo [11], Ethiopia [12], Gabon [13], Ghana [14], Kenya [15,16], Madagascar [17], Mali [18,19], Nigeria [20–24], Republic of the Congo [25], Rwanda [26], Senegal [18], Sudan [27], Tanzania [28], The Gambia [18], and Zambia [29,30]); 13 countries in the Americas (Bolivia [1], Brazil, Colombia, French Guiana (see references in S2 Table), Guatemala [31], Honduras [32], Jamaica [33], Martinique [34], Mexico [32,35], Nicaragua (references in S2 Table), Peru [36], Puerto Rico [32], and Suriname [37]); 3 countries in the Eastern Mediterranean (Iran [38], Iraq [39], Saudi Arabia [40]); 3 countries in Europe (Cyprus [41], France [42] and Sweden [26]); 2 countries in Southeast Asia (Indonesia, Thailand (references in S2 Table); and 10 countries in the Western Pacific region (China [43–45], Fiji [46,47], French Polynesia (references in S2 Table), Lao PDR [48], Malaysia [49–51], Papua New Guinea [52], Solomon Islands [53], Taiwan [54], The Philippines [55], and Vietnam [56,57]). The majority of these studies were carried out within the tropical zone, with Brazil (n = 13) and Thailand (n = 7) being the most actively involved in survey implementation over the last century. Furthermore, more than 80.0% of the studies were published during and after 2019. The number of studies per country or territory is shown in Table 1 and Fig 2A.

The cross-sectional observational research design, which is considered the most suitable for measuring prevalence, was the most frequently employed (78/84, 92.9%). Sample sizes ranged widely, from 88 to 5,880 individuals per study, with a mean of 582.2 ±135.93. As for study populations, the majority focused exclusively on the general population (34/84, 40.5%), while 19.0% centered on pregnant women, and 16.7% on blood donors. Among the eighty-three studies reporting age, 61.4% targeted adults, 32.6% encompassed all age groups, and 6.0% exclusively studied children. In terms of study settings, data were available for 67 studies. Among these, 64.2% were conducted in urban areas, 17.9% in both urban and rural settings, 11.9% in rural areas and the remaining studies in semi-urban or peri-urban. Furthermore, females accounted for 59.2% of the total participants.

The risk of bias was categorized as low in 76.2% and moderate in 23.8% of articles. The most significant issue in the evaluation was the absence of national population representation (S3 Table).

## Seroprevalence methods

Methods employed for the detection of IgG antibodies against ZIKV involved a wide range of techniques. These methods included both in-house and commercial enzyme-linked immunosorbent assay (ELISA), with a predominant usage of the commercial indirect Euroimmun-NS1 based kit. Other techniques included lateral flow immunoassays (LFA), blockade-of-binding assay (BOB), indirect immunofluorescent test (IIFT), colorimetric assays, Luminex bead-based assay, hemagglutination inhibition assay (HIA), microsphere immunoassay (MIA), and a variety of neutralization tests, such as plaque reduction neutralization tests (PRNT 50 & 90), focus reduction neutralization test (FRNT 50 & 90), and microneutralization assays (VNT).

For the interpretation of ZIKV seropositivity, ELISA tests were the most frequently used (34/83, 40.9%), followed by a combination of ELISA and a neutralization test (28/83, 33.7%),

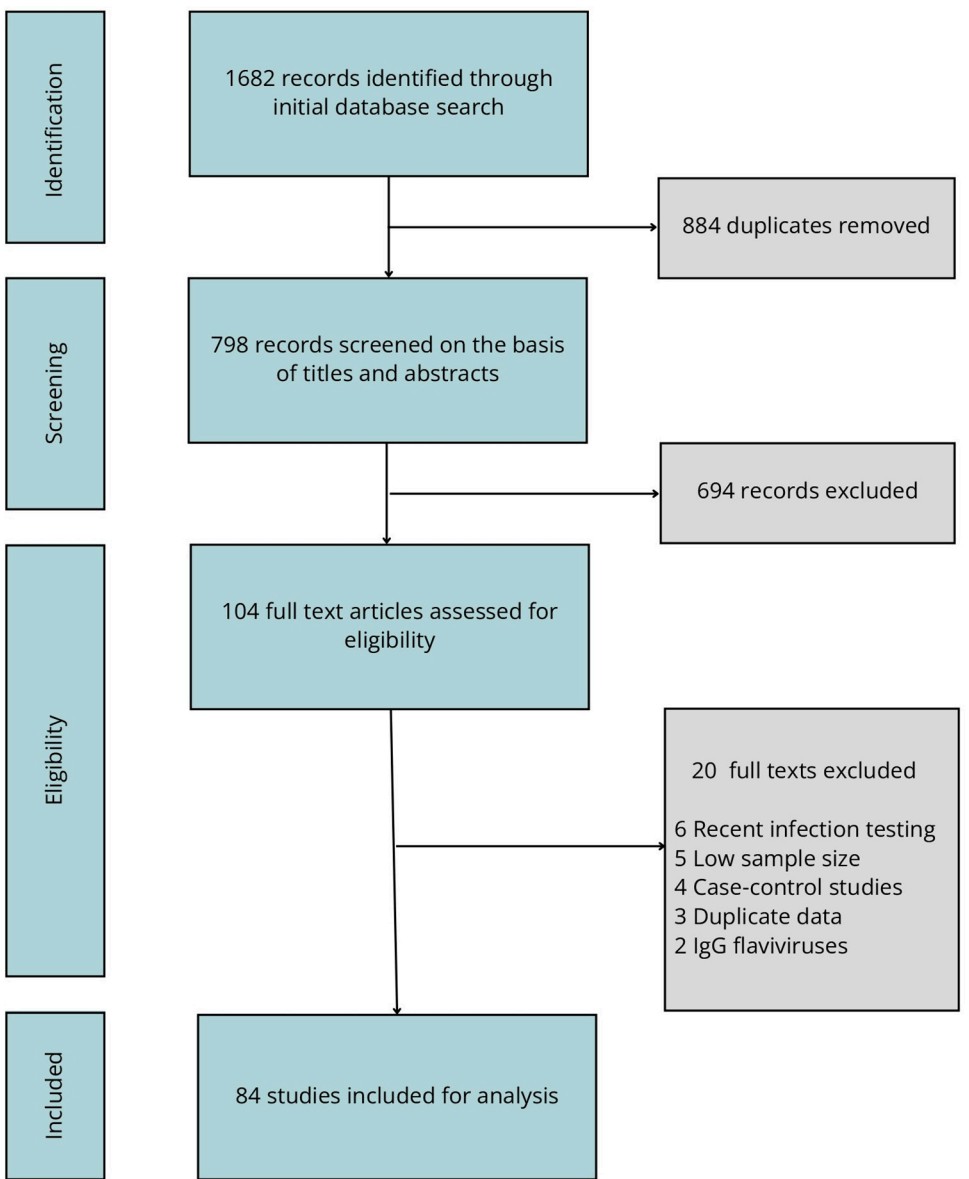

**Fig 1. Flowchart of the selection process of the studies included in the analysis of Zika virus seroprevalence.**

which included VNT (16/28, 57.1%), and well as PRNT or FRNT (12/28, 42.9%). In terms of ZIKV strains used for methodology, 36 out of 48 employed the Asian lineage, while 11 studies used the African lineage, and 1 Asian and African lineages.

## Seroprevalence rates

In the meta-analysis, substantial heterogeneity was observed both in the overall data and among subgroups. The global ZIKV seroprevalence, determined using a random-effects model, was 21.0% (95%CI 16.1%-26.4%, $I^2$ = 99.6%, p < 0.0001) (S1 Fig.). Egger's test yielded a p-value of 0.43, indicating no evidence of publication bias.

When dividing by WHO regions, the highest seroprevalence rate was observed in the Americas at 39.9% (95% CI: 30.3–49.9), followed by Southeast Asia at 22.8% (95% CI: 16.5–

**Table 1. Seroprevalence of Zika virus per World Health Organization region and per study period.**

| WHO region | Country | Studies per country | Study period | | | | | |
|---|---|---|---|---|---|---|---|---|
| | | No. | Jan 2000-Jul 2023 | | Jan 2000-Apr 2015 | | May 2015-Jul 2023 | |
| | | | Prevalence | | Prevalence | | Prevalence | |
| | | | % | 95%CI | % | 95%CI | % | 95%CI |
| Worldwide (84 studies) | Overall | 91[a] | 21.0 | 16.1–26.4 | 12.6 | 6.4–20.4 | 25.7 | 20.5–31.3 |
| Africa | All | 25 | 8.4 | 4.8–12.9 | 5.1 | 1.2–11.2 | 15.0 | 9.2–21.9 |
| | Burkina Faso | 1 | 22.8 | 19.2–26.5 | NA | NA | 22.8 | 19.2–26.5 |
| | Cabo Verde | 1 | 10.9 | 8.1–14.0 | NA | NA | 10.9 | 8.1–14.0 |
| | Cameroon | 1 | 4.9 | 3.7–6.3 | NA | NA | 4.9 | 3.7–6.3 |
| | Democratic Republic of the Congo | 1 | 0.1 | 0.0–0.4 | 0.1 | 0.0–0.4 | NA | NA |
| | Ethiopia | 1 | 27.3 | 20.5–34.8 | NA | NA | 27.3 | 20.5–34.8 |
| | Gabon | 1 | 43.7 | 33.7–53.9 | 52.6 | 36.6–68.4 | 41.6 | 30.4–53.1 |
| | Ghana | 1 | 13.1 | 9.1–17.8 | 13.1 | 9.1–17.8 | NA | NA |
| | Kenya | 2 | 1.5 | 0.0–7.1 | 0.2 | 0.0–0.7 | 3.9 | 2.7–5.3 |
| | Madagascar | 1 | 0.0 | NA | 0.0 | NA | NA | NA |
| | Mali | 2 | 7.0 | 1.7–15.6 | 5.6 | 3.0–16.5 | 12.0 | 9.8–14.3 |
| | Nigeria | 5 | 13.1 | 6.6–21.3 | NA | NA | 13.1 | 6.6–21.3 |
| | Republic of the Congo | 1 | 1.8 | 0.7–3.4 | 1.8 | 0.7–3.4 | NA | NA |
| | Rwanda | 1 | 1.4 | 0.7–2.3 | NA | NA | 1.4 | 0.7–2.3 |
| | Senegal | 1 | 10.1 | 4.8–17.0 | 10.1 | 4.8–17.0 | NA | NA |
| | Sudan | 1 | 0.1 | 0.0–0.2 | 0.1 | 0.0–0.2 | NA | NA |
| | Tanzania | 1 | 6.7 | 5.5–7.9 | NA | NA | 6.7 | 5.5–7.9 |
| | The Gambia | 1 | 1.8 | 0.0–6.2 | 1.8 | 0.0–6.2 | NA | NA |
| | Zambia | 2[b] | 7.9 | 3.9–13.1 | NA | NA | 10.7 | 6.9–15.3 |
| America | All | 31 | 39.9 | 30.3–49.9 | 0.0 | NA | 39.9 | 30.3–49.9 |
| | Bolivia | 1 | 29.1 | 26.0–32.3 | NA | NA | 29.1 | 26.0–32.3 |
| | Brazil | 13 | 36.9 | 24.9–49.8 | 0.0 | NA | 36.9 | 24.9–49.8 |
| | Colombia | 4 | 56.6 | 15.7–92.7 | NA | NA | 56.6 | 15.7–92.7 |
| | French Guiana | 1 | 23.3 | 21.7–24.9 | NA | NA | 23.3 | 21.7–24.9 |
| | Guatemala | 1 | 21.8 | 17.8–26.1 | NA | NA | 21.8 | 17.8–26.1 |
| | Honduras | 1 | 73.8 | 69.9–77.6 | NA | NA | 73.8 | 69.9–77.6 |
| | Jamaica | 1 | 15.6 | 12.7–18.6 | NA | NA | 15.6 | 12.7–18.6 |
| | Martinique | 1 | 21.9 | 18.6–25.3 | NA | NA | 21.9 | 18.6–25.3 |
| | Mexico | 2 | 60.1 | 56.9–63.2 | NA | NA | 60.1 | 56.9–63.2 |
| | Nicaragua | 3 | 57.0 | 37.8–75.1 | NA | NA | 57.0 | 37.8–75.1 |
| | Peru | 1 | 13.0 | 9.9–16.5 | NA | NA | 13.0 | 9.9–16.5 |
| | Puerto Rico | 1 | 34.0 | 29.4–38.7 | NA | NA | 34.0 | 29.4–38.7 |
| | Suriname | 1 | 35.1 | 31.7–38.5 | NA | NA | 35.1 | 31.7–38.5 |
| Eastern Mediterranean | All | 3 | 16.0 | 10.1–23.1 | NA | NA | 16.0 | 10.1–23.1 |
| | Iran | 1 | 0.0 | NA | NA | NA | 0.0 | NA |
| | Iraq | 1 | 12.0 | 6.5–18.9 | NA | NA | 12.0 | 6.5–18.9 |
| | Saudi Arabia | 1 | 18.5 | 14.9–22.5 | NA | NA | 18.5 | 14.9–22.5 |
| Europe | All | 3 | 0.0 | NA | NA | NA | 0.0 | NA |
| | Cyprus | 1 | 0.0 | NA | NA | NA | 0.0 | NA |
| | France | 1 | 0.0 | NA | NA | NA | 0.0 | NA |
| | Sweden | 1 | 0.0 | NA | NA | NA | 0.0 | NA |

(*Continued*)

**Table 1.** (Continued)

| WHO region | Country | Studies per country | Study period | | | | | |
|---|---|---|---|---|---|---|---|---|
| | | No. | Jan 2000-Jul 2023 | | Jan 2000-Apr 2015 | | May 2015-Jul 2023 | |
| | | | Prevalence | | Prevalence | | Prevalence | |
| | | | % | 95%CI | % | 95%CI | % | 95%CI |
| Southeast Asia | All | 10 | 22.8 | 16.5–29.7 | 23.3 | 13.6–34.5 | 22.1 | 16.4–28.4 |
| | Indonesia | 3 | 9.1 | 7.7–10.6 | 9.1 | 7.7–10.6 | 0.0 | NA |
| | Thailand | 7 | 25.9 | 19.5–32.8 | 29.3 | 18.7–41.1 | 22.1 | 16.4–28.4 |
| Western Pacific | All | 19 | 15.6 | 8.2–24.9 | 16.1 | 3.0–36.5 | 15.5 | 6.8–26.8 |
| | China | 3 | 2.1 | 0.0–11.9 | NA | NA | 2.1 | 0.0–11.9 |
| | Fiji | 2 | 13.7 | 7.4–21.4 | 7.5 | 5.9–9.3 | 19.2 | 12.5–26.9 |
| | French Polynesia | 4 | 40.7 | 26.6–67.4 | 40.0 | 8.2–77.5 | 42.1 | 7.5–82.0 |
| | Lao PDR | 1 | 7.0 | 2.7–13.2 | 4.5 | 2.5–6.9 | 9.9 | 7.8–12.2 |
| | Malaysia | 3 | 7.1 | 0.7–19.0 | 1.7 | 0.2–4.2 | 7.7 | 0.0–37.8 |
| | Papua New Guinea | 1 | 64.9 | 58.3–71.3 | NA | NA | 64.9 | 58.3–71.3 |
| | Solomon Islands | 1 | 28.1 | 25.4–30.9 | NA | NA | 28.1 | 25.4–30.9 |
| | Taiwan | 1 | 0.5 | 0.0–2.0 | NA | NA | 0.5 | 0.0–2.0 |
| | The Philippines | 1 | 17.9 | 14.8–21.2 | NA | NA | 17.9 | 14.8–21.2 |
| | Vietnam | 2 | 1.0 | 0.4–1.7 | 0.6 | 0.0–2.4 | 1.1 | 0.5–2.0 |

NA: No data available.

[a]84 studies in total: 4 studies including more than one country.

[b]One of the studies without data for the year

29.7). Lower seroprevalence rates were observed in the Eastern Mediterranean (16.0%, 95% CI: 10.1–23.1), Western Pacific (15.6%, 95% CI: 8.2–24.9), Africa (8.4%, 95% CI: 4.8–12.9) and Europe (0.0%).

Comparing seroprevalences between two time periods (Jan 2000 to Apr 2015 and May 2015 to Jul 2023), the overall seroprevalence increased from 12.6% (95%CI 6.4–20.4%) to 25.7% (95%CI 20.5–31.3%). Evidence of ZIKV circulation during both periods was documented in Africa (5.1% vs 15.0%), Southeast Asia (23.3% vs 22.1%), and Western Pacific (16.1% vs 15.5%). Further details on heterogeneity among studies, seroprevalence rates across WHO regions, countries, and time periods are available in Tables 1, 2, S4 and S5.

The top five countries with the highest seroprevalence rates were Honduras (73.8%, 95%CI 69.9–77.6%), Papua New Guinea (64.9%, 95%CI 58.3–71.3%), Mexico (60.1%, 95%CI 56.9–63.2%), Nicaragua (57.0%, 95%CI 37.8–75.1%), and Gabon (43.7%, 95%CI 33.7–53.9%). Conversely, the countries with the lowest ZIKV seroprevalence rates were Madagascar, Iran, and European countries (0.0%). Fig 2B displays seroprevalence rates by country.

Before May 2015, provinces like Moyen-Ogooué in Gabon, Ratchaburi in Thailand (~55.0%), and the Society Islands in French Polynesia (40.0%), exhibited high seropositivity, determined through ELISA tests. After 2015, high seroprevalences rates were observed in various regions. In Africa: Moyen-Ogooué province in Gabon (maintained a high seroprevalence 41.6%), followed by Gambella, Ethiopia (27.3%, BOB ELISA, African strain), and Ouagadougou and Bobo-Dioulasso, Burkina Faso (22.8%, cELISA & VNT, African strain) [8]. In the Americas: Departments of Córdoba (89.0%, ELISA) and Risaralda (86.8%, ELISA & FRNT) in Colombia, Acapulco in Mexico, and Tegucigalpa in Honduras (85.0%, 73.8%, respectively; BOB ELISA and VNT). In the Eastern Mediterranean: The cities of Najran (18.5%) in Saudi Arabia, and Basrah in Iraq (12.0%) both determined using ELISA tests. In Southeast Asia:

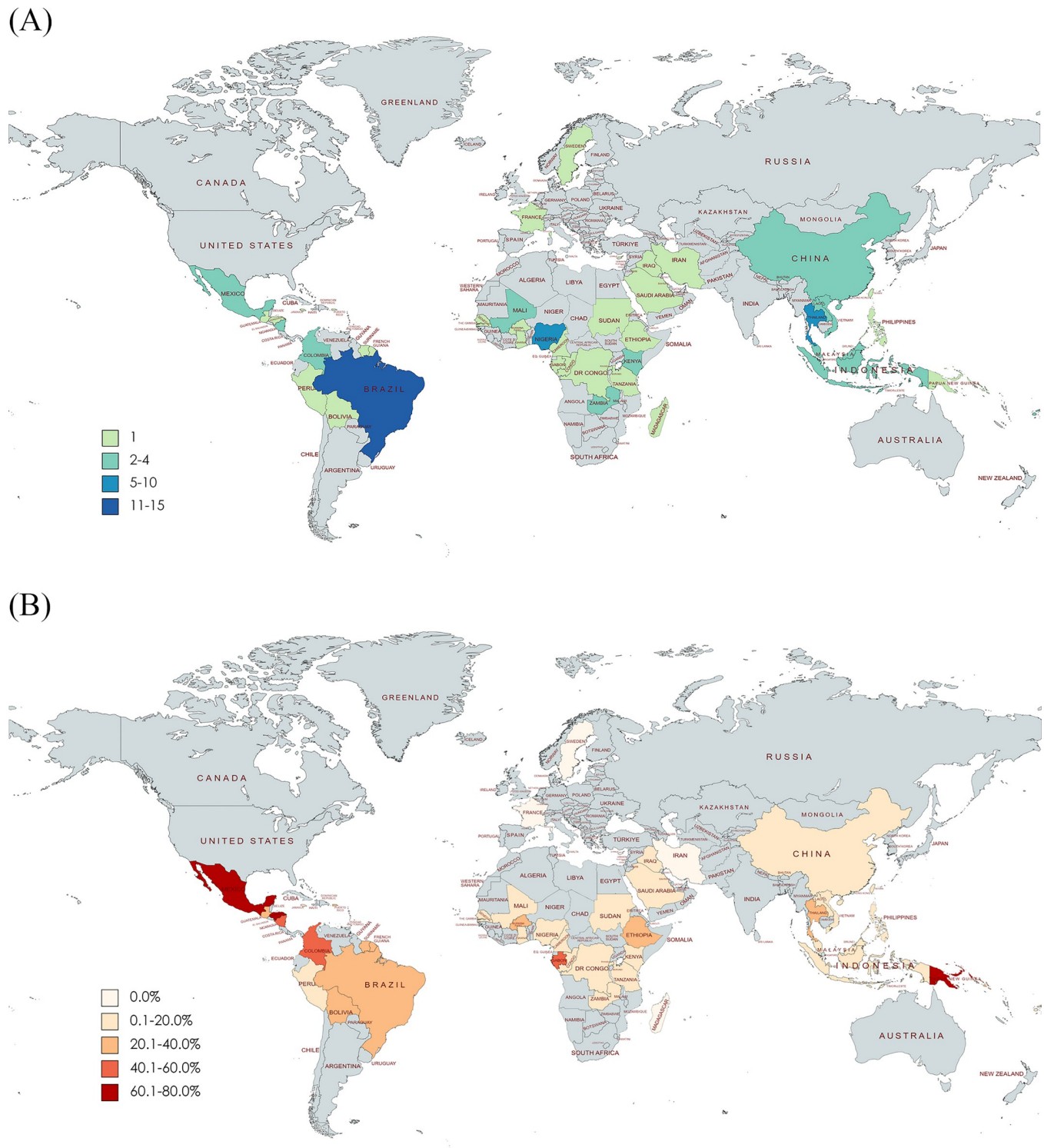

**Fig 2. (A) Number of Zika virus seroprevalence studies per country. (B) Seroprevalence of Zika virus per country.** The maps were created using Mapchart. World Map—Simple | Create a custom map. In: MapChart [Internet]. [cited 5 Mar 2024]. Available: https://mapchart.net/world.html.

**Table 2. Subgroup analysis of Zika virus seroprevalence per World Health Organization region.**

| Characteristics | Worldwide | | Africa | | America | | Eastern Mediterranean | | Southeast Asia | | Western Pacific | |
|---|---|---|---|---|---|---|---|---|---|---|---|---|
| | Prevalence | | Prevalence | | Prevalence | | Prevalence | | Prevalence | | Prevalence | |
| | % | 95%CI | % | 95%CI | % | 95%CI | % | 95%CI | % | 95%CI | % | 95%CI |
| **Population type** | | | | | | | | | | | | |
| General | 23.4 | 17.2–30.2 | 16.1 | 7.0–27.8 | 42.8 | 32.4–53.4 | 12.0[a] | 6.5–18.9 | 28.7 | 14.9–44.9 | 15.2 | 6.9–25.9 |
| Pregnant women | 32.2 | 21.0–44.5 | 11.9 | 5.5–20.4 | 62.3 | 44.4–78.5 | 18.5[a] | 14.9–22.5 | 25.1 | 21.4–28.9 | 1.0 | 0.4–1.8 |
| Blood donors | 8.2 | 3.2–17.7 | 6.1 | 1.3–14.0 | 10.4 | 3.5–20.3 | NA | NA | NA | NA | 2.9 | 0.6–6.6 |
| **Age** | | | | | | | | | | | | |
| Children | 14.3 | 6.9–23.8 | 4.9 | 0.05–17.1 | 27.3 | 20.9–34.1 | NA | NA | 9.2 | 7.9–10.8 | 16.4 | 13.4–19.8 |
| Adults | 20.6 | 15.2–26.6 | 8.4 | 4.7–5.5 | 39.8 | 32.0–47.9 | 18.5[a] | 14.9–22.5 | 24.1 | 20.3–28.2 | 9.2 | 3.2–17.8 |
| Children and Adults[b] | 15.5 | 7.0–25.7 | 6.3 | 5.6–6.9 | 37.6 | 8.9–72.3 | 3.5 | 2.0–24.6 | 7.7 | 0.5–35.7 | 25.3 | 9.6–45.4 |
| **Settings** | | | | | | | | | | | | |
| Urban | 19.7 | 13.8–26.4 | 11.1 | 5.3–18.7 | 38.0 | 27.7–49.0 | 17.0 | 13.9–20.4 | 23.3 | 12.4–36.4 | 4.3 | 1.3–8.8 |
| Rural | 14.1 | 7.3–22.7 | 7.2 | 3.1–12.7 | 28.6 | 6.9–57.5 | 12.0[a] | 6.5–18.9 | NA | NA | 9.1 | 2.9–18.3 |
| Semi-urban | 31.0 | 14.9–50.0 | 43.7[a] | 33.7–53.9 | NA | NA | NA | NA | NA | NA | 17.5 | 15.0–20.2 |
| **Serological methods** | | | | | | | | | | | | |
| ELISA | 30.6 | 22.9–38.9 | 19.5 | 9.4–32.0 | 43.2 | 27.6–59.4 | 16.0 | 10.1–23.1 | 26.8 | 17.2–37.7 | 34.0 | 12.4–59.8 |
| ELISA & PRNT or FRNT | 14.7 | 3.6–31.4 | 0.1 | 0.0–0.2 | 39.5 | 10.6–73.3 | NA | NA | 30.8[a] | 27.3–34.4 | 6.6 | 0.4–19.0 |
| ELISA & VNT | 15.8 | 9.5–23.4 | 6.6 | 3.2–11.1 | 35.0 | 22.8–48.2 | NA | NA | NA | NA | 4.8 | 1.5–9.6 |
| PRNT90 | 12.9 | 5.7–22.5 | 3.9[a] | 2.7–5.3 | NA | NA | NA | NA | 15.9 | 8.0–25.8 | 0.0[a] | NA |
| MIA | 20.8 | 9.1–35.7 | NA | NA | NA | NA | NA | NA | NA | NA | 20.8 | 9.1–35.7 |
| Other methods | 22.1 | 10.8–36.1 | 16.1 | 9.2–24.2 | 39.6 | 28.5–51.2 | NA | NA | 17.0[a] | 11.3–23.5 | 8.7[a] | 0.0–43.2 |
| **ZIKV strain** | | | | | | | | | | | | |
| Asian | 24.9 | 17.9–32.6 | 4.1 | 0.9–9.5 | 44.9 | 34.9–55.2 | NA | NA | 13.1 | 8.0–19.1 | 17.0 | 8.5–27.6 |
| African | 9.1 | 3.1–17.6 | 6.1 | 2.1–11.8 | NA | NA | NA | NA | 26.7[a] | 24.5–28.9 | 2.5 | 0.0–94.6 |

[a]Less than two studies

[b]No individual data. Europe: No seropositivity

Abbreviations: ELISA: Enzyme-linked immunosorbent assay; FRNT: Focus reduction neutralization assay; MIA: Microsphere immunoassay; NA: No data available; PRNT: Plaque reduction neutralization test; VNT: Virus neutralization test.

Samut Prakan province in Thailand (45.4%, ELISA). In the Western Pacific region: Manus and Wewak Islands (~65.0%) in Papua New Guinea (ELISA and VNT, African strain), North Guadalcanal and Honiara (~55%) in the Solomon Islands (ELISA), and the Society Islands in French Polynesia (42.1%, similar seroprevalence rates as before 2015).

The lowest seroprevalence rates after the emergence of ZIKV in the Americas were found in the highlands of Bolivia (La Paz and Cochabamba, 0.0%) [1], the subtropics of Brazil (Santa Maria, Rio Grande do Sul, 0.6%), China (Guizhou province, 0.0%), and Taiwan (Tainan, 0.5%), and in the tropical rainforest of Indonesia (Aceh province, 0.0%). The seroprevalence rates by administrative division and by sampling period are shown in Fig 3A and 3B.

In subgroup analysis, the seroprevalence was estimated to be higher in pregnant women (32.2%, 95%CI 21.0–44.5%), than in the general population (23.4%, 95%CI 17.2–30.2) and in blood donors (8.2%, 95%CI 3.2–17.7) (p<0.001). Globally, adults were significantly more exposed to ZIKV compared to exclusively children (20.6% vs 14.3%, p-value <0.001), with particularly low seroprevalence in children in Africa (4.9%) and Southeast Asia (9.2%). Seroprevalence was higher in semiurban (31.0%, 14.9–50.0%) and urban (19.7%, 13.8–26.4%) compared

(A)

(B)

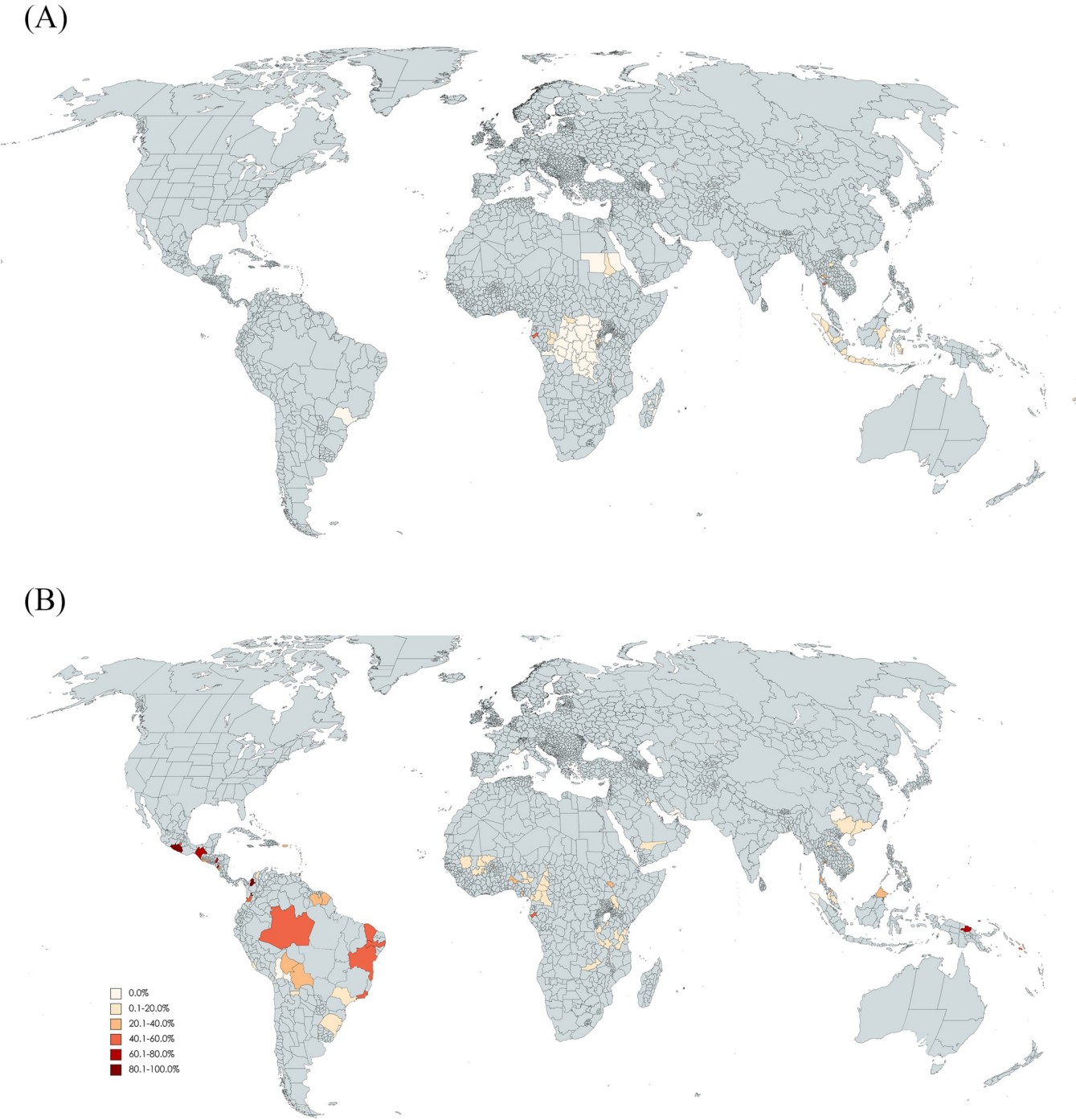

**Fig 3. (A) Seroprevalence of Zika virus according to administrative divisions from January 2000 to April 2015. (B) Seroprevalence of Zika virus according to administrative divisions from May 2015 to July 2023.** The maps were created using Mapchart. World Map—Advanced | MapChart. [cited 5 Mar 2024]. Available: https://www.mapchart.net/world-advanced.html.

to rural areas (14.1%, 7.3–22.7%). Moreover, increased sensitivity of ELISA tests (30.6%, 22.9–38.9%) was significantly associated with increased seroprevalence compared to the combination of ELISA and a seroneutralization tests (PRNT or FRNT 14.7%, 3.6–31.4%, or VNT

15.8%, 9.5–23.4%, p-value <0.001). Using the Asian lineage (24.9%, 17.9–32.6%), resulted in higher seroprevalence compared to the African lineage (9.1%, 3.1–17.6%) (p-value <0.001). Additionally, other factors contributing to seropositivity in specific studies included low educational levels, low socio-economic status, and polygamy.

## Discussion

Zika virus is a mosquito-borne illness that has affected thousands of people in tropical and subtropical regions worldwide and has caused severe birth effects. In this systematic review and meta-analysis, we retrieved information on the prevalence of ZIKV infection from January 2000 to July 2023 and analyzed it at global, regional, and national levels. Our study aimed to understand and provide a more comprehensive perspective of global ZIKV infection, identifying trends over time, and addressing future challenges.

The estimated pooled global seroprevalence was 21.0% by involving 84 studies and 63,864 individuals. The findings indicate a high heterogeneity of study designs and serological tests. Similar seroprevalence was identified in a previous systematic review analyzing 12 studies until 2019 [6] and indicate a progressive growing of ZIKV seroprevalence studies in the last four years.

The assessment of seroprevalence involved a variety of diagnostic methods, resulting in challenges when determining the relative accuracy, sensitivity, and specificity, as well as when estimating the performance value of each study through proper comparison. The most commonly used method, the indirect semi-quantitative NS1 ELISA IgG, was preferred due to its higher sensitivity, stronger response, longer duration of detectability, and reduced cross-reactivity compared to the envelope protein E, which is highly conserved and leads to suboptimal diagnoses [58]. However, significant cross-reactivity persists, as demonstrated in a study in Brazil with 54.0% of cross-reactivity when compared with PRNT [59], and 60.0% in a study in Bolivia compared with VNT [1]. Nonetheless, significant advancements and investments have been made in the development and improvement of tests. For instance, the use of the WHO International Standard (WHO ISs) for anti-Asian linage ZIKV antibody (IS 16_352) allows for the comparison of biological activity by defining an internationally agreed unit, the International Unit (IU27), using a calibrator for serology assays. This standardization facilitates the harmonization of data from different laboratories worldwide and provides better definition of parameters such as the analytical sensitivity of tests and protective levels of antibody [60]. Other promising tests such as the ZIKV NS1 BOB ELISA, a simple, robust, and cost-effective test, have been used in some seroprevalence studies. In this assay, serum antibodies are measured for their ability to block the binding of a ZIKV NS1- specific monoclonal antibody to solid-phase ZIKV NS1. This method has shown higher specificity compared to other ELISAs, but should undergo further validation [41]. Additionally, some studies have used the envelope protein domain III (EDIII), which is highly diverse among flaviviruses and can improve sensitivity and specificity of the assays [61]. Nevertheless, neutralization tests continue to be considered the "gold standard" and provide better specificity, but the potential for cross-reactivity persists in flavivirus-endemic areas. Variability across neutralization tests, including the type of assay, target cells, virus stock, timings, and titer threshold, has been observed. According to the WHO guidelines for recent ZIKV infection, a PRNT using a 90% cutoff value with a titer ≥10 and negativity for other flaviviruses is confirmatory [62]. However, no information regarding the titer in seroprevalence studies is available. The implementation of standardized protocols is essential, as they will not only enable the comparison of results across different regions and countries but also enhance the quality of studies by reducing potential biases.

In Africa, ZIKV has been circulating since 1945 [2], primarily the African genotype. However, large-scale outbreaks have been relatively infrequent, as shown in our study, with an overall seroprevalence of 8.4%, which increased from 5.1% before May 2015 to 15.0% after May 2015. According to the WHO, ZIKV transmission has been reported in 14 African countries or territories, including those where seroprevalence studies were conducted, such as Burkina Faso (with a seroprevalence of 22.8%), Cabo Verde (10.9%), Cameroon (4.9%), Ethiopia (27.3%), Gabon (43.7%), Kenya (1.5%), Nigeria (13.1%), and Senegal (10.1%) [63]. The high seroprevalence found in Gabon is attributable to the largest epidemic ever recorded in Africa, which occurred in 2007. This epidemic was potentially caused by *Ae. albopictus*, which is present in all human environments in Gabon and is thought to be more susceptible to African ZIKV [64] compared to African *Ae. aegypti*, which is more resistant to American and Asian ZIKV. However, an exception is the emergence of the Asian lineage in Cabo Verde (7,580 suspected cases) in 2015–2016 caused by *Ae. aegypti* [65]. Studies indicate that *Ae. aegypti* in Cabo Verde appear to have a strong attraction to human blood and a high susceptibility to ZIKV infection compared to generalist ancestral subspecies *Ae. aegypti formosus* (Aaf), which predominates in Africa. This genomic profile was also identified in Senegal, which had a similar seroprevalence between 2007 and 2012, and in Angola, where only four acute cases were detected in 2016–2017 [66]. Moreover, the moderate seroprevalence found in Ethiopia is associated with individuals who visited forest areas [12]. As for Burkina Faso, further studies are needed to understand and potentially identify any undetected epidemics [8]. There is a high public health concern regarding the possibility of African ZIKV triggering large urban epidemics, along with the risk associated with *Ae. albopictus*. Experiments have demonstrated that African strains are more pathogenic than Asian strains, and that *Ae. aegypti* mosquitoes are highly competent to transmit the different genotypes, specially, the African [64]. Hence, there is a need to enhance vector surveillance and control methods in the region to reduce the risk of potential future outbreaks.

In the Americas, autochthonous vector-borne transmission of ZIKV has been confirmed in all countries and territories, except for continental Bermuda, Canada, mainland Chile, and Uruguay. Regarding the number of cases reported between 2015 and 2023, Brazil (4th in incidence), Colombia and Venezuela top the list with 508,609, 111,744 and 62,093 cases, respectively [67]. Our meta-analysis revealed a seroprevalence of 39.9% within the region, which ascends to ~50% in many tropical regions, reflecting the substantial degree of exposure to ZIKV among the population, particularly in Honduras, Mexico, and Colombia. However, comparing the number of reported cases by country and their seroprevalence presents challenges. On one hand, there are disparities in case reporting, such as confirmed cases (e.g. Mexico) versus probable and confirmed cases (e.g. Brazil, Colombia). In addition to variations in diagnostic protocols and laboratory capacity. On the other hand, seroprevalence is influenced by variations in methods, study periods, the number of studies, and study regions. Consequently, these limitations may affect the generalizability of the findings.

Currently, there is a growing concern in the region due to the recent confirmation of the presence of the *Ae. aegypti* in the highlands of Bolivia (Cochabamba, 2,558 m), the expansion of the vector's distribution in new regions of Chile (e.g. Parinacota and Los Andes in Valparaíso Region), the identification of local Chikungunya transmission in Uruguay in 2023 [68], and a notable increase in arbovirus activity in the state of Rio Grande do Sul, Southern Brazil [69]. These expansions are attributable to climate change, heightening the potential for future transmission in previously unaffected areas. Additional risks involve the recent identification of the African lineage in Brazil among non-human primates and mosquitoes in geographically and climatically distinct regions (South and Southeast), posing a potential threat associated with birth defects [70]. Therefore, further research is essential to better

understand the prevalence of ZIKV, vector dynamics, and the genetic evolution of the virus in the Americas.

In the Eastern Mediterranean Region (EMR), no country has reported autochthonous transmission of ZIKV. Nevertheless, documented records indicate the presence of *Ae. aegypti* populations in Afghanistan, Djibouti, Egypt, Oman, Pakistan, Saudi Arabia, Somalia, Sudan, and Yemen [4]. Seroprevalence studies have revealed the presence of IgG antibodies in two countries within their population. First, in the city of Najran in southwestern Saudi Arabia, with a seroprevalence of 18.5% between 2016–2017. This region experiences a growing annual increase in dengue cases [71]. Second, in the city of Basra, southern Iraq, with a seroprevalence of 12.0% in 2019–2020. This country is free of transmission and presence of known vectors. However, these studies were performed using ELISA tests. Consequently, it is challenging to discern whether this seropositivity is the result of possible cross-reaction with other flaviviruses or actual ZIKV transmission. A study has demonstrated that all EMR countries are suitable for *Ae. aegypti* and *Ae. albopictus*. Thus, it is imperative to undertake active entomological and epidemiological surveillance throughout the region to help identify the introduction of mosquito species and the circulation of arboviruses [72].

In Europe, Southern France (Hyeres, Var department) was the first and only region reporting three autochthonous transmission in October 2019. Subsequently, a seroprevalence study was undertaken in the same area, but it yielded no positive cases [42]. This localized transmission event was attributed to *Ae. albopictus*, a mosquito species well-established in France since 2004, potentially by the Asian genotype originating from Southeast Asia [73]. Previous studies revealed the reduced susceptibility of the French *Ae. albopictus* populations (Montpellier and Corsica) to the Asian ZIKV compared to the African ZIKV genotype. Moreover, these mosquitoes demonstrated limited capacity to sustain local virus transmission [74]. It is plausible that environmental factors played a role in this brief episode of local transmission, despite the apparent limitations in vector competence [73]. To gain a deeper understanding of the potential for ZIKV to established a sustained transmission cycle involving *Ae. albopictus* in southern Europe, further investigations are needed. This is especially crucial in light of the evolving epidemiological situation of Dengue virus in France, which has seen a rising number of autochthonous cases being reported recently [75].

In Southeast Asia, ZIKV has been circulating since at least the 1950s. Surprisingly, ZIKV outbreaks in this region have remained moderate in intensity, with notable outbreaks occurring in Thailand in 2016–2018 (n = 2,300), and in India in 2018 (n = 283 cases) [2,76]. In our study, the estimated seroprevalence was determined to be 25.9% (Thailand 25.9%, Indonesia 9.1% in children), with evidence of prior circulation before May 2015. Studies have hypothesized that the observed differences in ZIKV outbreaks patterns between Southeast Asia and the Americas may be attributed to the fact that, despite the highly efficient *Ae. aegypti* mosquitoes in Southeast Asia, the transmission efficiency of Asian ZIKV strains may not be as robust as that of American strains [77], and/or to the acquisition of genetic changes that increased ZIKV infectivity and prevalence in mosquitoes of the Americas [78]. Other studies also indicate that in regions with endemic transmission, such as Thailand, there is an increase in genetic diversity among circulating lineages that would limit sustained transmission chains [79]. To gain deeper insights into the genetic diversity and the persistent circulation in the region, it is imperative to conduct extensive surveillance in competent mosquito vectors across Southeast Asia. Moreover, Thailand's laboratory capabilities should be shared to improve the capacities of neighboring countries in the region.

In the Western Pacific region, large epidemics, likely caused by the American sublineage, were reported in French Polynesia between 2013 and 2014, followed by outbreaks in the South Pacific Islands [3]. Correspondingly, studies showed higher seroprevalence in Fiji (13.7%),

French Polynesia (40.7%), Papua New Guinea (64.9%), and Solomon Islands (28.1%). Conversely, sporadic cases or small outbreaks, probably caused by the Asian sublineage [3], were identified in countries where seroprevalence studies were conducted, such as Lao PDR (7.0%), Malaysia (7.1%), The Philippines (17.9%), and Vietnam (1.0%) [2,80]. Consequently, Malaysia, Laos and Vietnam may have lower levels of immunity, making them highly vulnerability to future epidemics. Moreover, evidence indicating a decline in seroprevalence over a two-year period reported in French Polynesia [46] underscores the need for exhaustive monitoring of ZIKV.

In subgroup analysis, higher seroprevalence was identified in: a) Pregnant women, possibly because some studies were conducted during epidemics and potentially involved a higher participation of pregnant women with previous symptoms, which might introduce biases into the results. b) Adults, known to result from greater exposure to outdoor activities, and longer exposure to the virus. c) Urban and semi-urban areas, linked to the urban maintenance cycles of ZIKV in recent large epidemics. Additionally, some studies suggest an association with low educational and low-socio-economic levels, likely resulting from poor-quality housing conditions, overcrowding, and the storage of water in containers that create breeding sites for mosquitoes. A study has also indicated an association with polygamous relationships, suggesting a possible transmission through sexual contact.

Our meta-analysis faced several limitations. Our literature search highlights the highly heterogeneous seroprevalence of ZIKV worldwide. Some of these variations may stem from methodological differences, as well as the choice of study population and sample size. This heterogeneity may also reflect differential exposure to mosquitoes and the period of study—before, during, or after the end of the outbreak. Other limitations include the differences in the number of studies per country, which may increase or decrease the seroprevalence. Additionally, our study was based only in the detection of IgG or IgM/IgG. Although this may represent a major limitation for the comparison of seroprevalence, this study has strengths. A comprehensive search strategy was performed, with almost 80.0% of the studies assessed as having low risk of bias in their methodological quality. This suggests that we can be confident in the quality of our findings, which provide an updated global picture of the Zika situation so far.

In conclusion, ZIKV infection remains a public health threat. As ongoing social, environmental and economic changes continue to alter our environment, arboviral importation and transmission will persist. Addressing remaining gaps in its distribution requires the implementation of standardized seroprevalence studies, including the development of more sensitive and specific diagnostic tools.

## Supporting information

**S1 Table. Prisma checklist.**
(PDF)

**S2 Table. Characteristics of Zika virus seroprevalence studies included in the systematic review and meta-analysis.**
(PDF)

**S3 Table. Risk of bias assessment.**
(PDF)

**S4 Table. Heterogeneity assessment of seroprevalence of Zika virus per World Health Organization region and per study period.**
(PDF)

**S5 Table. Heterogeneity assessment of subgroup analysis of Zika virus seroprevalence per World Health Organization region.**
(PDF)

**S1 Fig. The global Zika virus seroprevalence using a random-effects model.**
(PDF)

## Author Contributions

**Conceptualization:** Paola Mariela Saba Villarroel, Sineewanlaya Wichit.

**Data curation:** Paola Mariela Saba Villarroel, Nuttamonpat Gumpangseth.

**Formal analysis:** Paola Mariela Saba Villarroel, Rodolphe Hamel, Sineewanlaya Wichit.

**Funding acquisition:** Sineewanlaya Wichit.

**Investigation:** Paola Mariela Saba Villarroel.

**Methodology:** Paola Mariela Saba Villarroel.

**Project administration:** Rodolphe Hamel, Dorothée Missé, Sineewanlaya Wichit.

**Supervision:** Sakda Yainoy, Phanit Koomhin.

**Validation:** Sakda Yainoy, Phanit Koomhin.

**Writing – original draft:** Paola Mariela Saba Villarroel.

**Writing – review & editing:** Paola Mariela Saba Villarroel, Rodolphe Hamel, Dorothée Missé, Sineewanlaya Wichit.

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
