## [Decision Letter · Decision Letter 0]

5 Mar 2024

Dear Dr. Wichit,

Thank you very much for submitting your manuscript "Global seroprevalence of Zika virus in asymptomatic individuals: A systematic review" for consideration at PLOS Neglected Tropical Diseases. As with all papers reviewed by the journal, your manuscript was reviewed by members of the editorial board and by several independent reviewers. The reviewers appreciated the attention to an interesting topic. Based on the reviews, we are likely to accept this manuscript for publication, providing that you modify the manuscript according to the review recommendations. 

Sincerely,

Kebede Deribe, BSc, MPH, PhD

Guest Editor

Andrea Marzi

Section Editor

Reviewer's Responses to Questions

**Key Review Criteria Required for Acceptance?**

**Methods**

-Are the objectives of the study clearly articulated with a clear testable hypothesis stated?

-Is the study design appropriate to address the stated objectives?

-Is the population clearly described and appropriate for the hypothesis being tested?

-Is the sample size sufficient to ensure adequate power to address the hypothesis being tested?

-Were correct statistical analysis used to support conclusions?

-Are there concerns about ethical or regulatory requirements being met?

Reviewer #1: The well- defined and attainable objectives set for the study provided a solid foundation for a purposeful and successful research. The study design was appropriate and meticulous, defining every step in a very clear way. This study aimed to determine seroprevalence of Zika virus in a symptomatic patients globally, and included 84 studies which in theory seems to be a small number but this shows the need for more studies in this field. the statistical and ethical aspects of this study were satisfactory for the requirements.

Reviewer #2: The study’s objectives are clearly articulated, and the authors have meticulously designed their research to address the objectives. Their well-structured hypothesis allows for rigorous testing, ensuring that the study’s findings contribute meaningfully to the field. Overall, the research design aligns with the intended goals, fostering robust conclusions and advancing scientific understanding. The authors of the study have meticulously described the targeted population, ensuring that it aligns with the research hypothesis. Additionally, the sample size appears to be adequate for drawing meaningful conclusions. The statistical analyses employed by the authors are appropriate and contribute to supporting the study’s findings. Importantly, there are no ethical concerns raised by this research.

Reviewer #3: The methods are well described

**Results**

-Does the analysis presented match the analysis plan?

-Are the results clearly and completely presented?

-Are the figures (Tables, Images) of sufficient quality for clarity?

Reviewer #1: The study is well-regarded for its well-defined objectives, meticulous design, global relevance, robust statistical analysis, and a commitment to thorough reporting. The acknowledgement of the need for further research adds to the study's overall significance.

Reviewer #2: The analysis in the study closely aligns with the planned analysis, and the findings are effectively communicated. The tables and figures are also well-constructed, enhancing the clarity of the research.

Reviewer #3: The study are results are well presented

**Conclusions**

-Are the conclusions supported by the data presented?

-Are the limitations of analysis clearly described?

-Do the authors discuss how these data can be helpful to advance our understanding of the topic under study?

-Is public health relevance addressed?

Reviewer #1: The study's conclusion is strongly supported by the detailed data in the results section, aligning well with the study's goals. Additionally, the study is noteworthy for openly discussing its limitations, providing readers with a clear understanding of potential constraints and biases. This commitment to both strong evidence and transparent acknowledgment of limitations enhances the credibility and integrity of the research.

Reviewer #2: The authors have attempted to enhance our understanding of the global spread of the Zika virus and its associated public health concerns. However, the conclusion needs further expansion and is not well-written.

Reviewer #3: The study conclusions are well stated and supported by data

**Editorial and Data Presentation Modifications?**

Reviewer #1: (No Response)

Reviewer #2: Minor comments 

Abstract 

In the abstract part, no need any citation, the authors cited Hoy et al., 2012, better to remove it.

Introduction

Line 70: “The Eastern Mediterranean Region is the only exception”, this statement is not clear, Certainly! The author should elaborate further to provide clarity and ensure that their statement is easily understandable. Clear explanations are essential for effective communication.

Methods

The authors of the MS did not mention the Web of Science Core Collection or the Cochrane Library as databases for searching articles. Instead, they focused on other sources. However, it’s worth noting that these databases are widely used for academic research and could provide valuable information for future studies.

Why the authors excluded research involving travelers, migrants? As you know travelers and migrants play a crucial role in the transmission of Zika disease due to their mobility. When people move across borders or travel between regions, they can inadvertently carry the virus with them, potentially introducing it to new areas. This movement facilitates the spread of Zika and other infectious diseases globally. It’s essential to consider these population dynamics when addressing public health measures and disease prevention strategies.

Results 

Line 226 ZIKV strains used for methodology, 36 out of 48 employed the Asian lineage, while 11 studies

 used the African lineage, and 1 Asian and African lineages. The authors did not include phylogeny data in their analysis. However, they need to address how they compare the Asian lineage and African lineages of the Zika virus.

Reviewer #3: (No Response)

**Summary and General Comments**

Reviewer #1: In summary, the study is well-regarded for its clear and achievable goals, careful design, and global relevance in examining Zika virus seroprevalence. The inclusion of 84 studies suggests a need for more research. The commitment to thorough reporting and acknowledgment of limitations contributes to the overall trustworthiness of the research. The study's conclusion is supported by detailed results, emphasizing the importance of the findings.

Reviewer #2: This manuscript is commendable. Its organization is logical and coherent from section to section. The document is written and formatted to a professional level. However, there are minor revisions needed in the body of the text to address typographical errors and grammar issues.

Reviewer #3: The comments are given in a separate word file and within the manuscript pdf file.

PLOS authors have the option to publish the peer review history of their article (what does this mean?). If published, this will include your full peer review and any attached files.

Reviewer #1: Yes: Rowa Hassan

Reviewer #2: No

Reviewer #3: Yes: Collins Okoyo

Figure Files:

Data Requirements:

Reproducibility:

References

---

## [Decision Letter · Decision Letter 1]

1 Apr 2024

Dear Dr. Wichit,

We are pleased to inform you that your manuscript 'Global seroprevalence of Zika virus in asymptomatic individuals: A systematic review' has been provisionally accepted for publication in PLOS Neglected Tropical Diseases.

Best regards,

Kebede Deribe, BSc, MPH, PhD

Guest Editor

Andrea Marzi

Section Editor

Reviewer's Responses to Questions

**Key Review Criteria Required for Acceptance?**

**Methods**

-Are the objectives of the study clearly articulated with a clear testable hypothesis stated?

-Is the study design appropriate to address the stated objectives?

-Is the population clearly described and appropriate for the hypothesis being tested?

-Is the sample size sufficient to ensure adequate power to address the hypothesis being tested?

-Were correct statistical analysis used to support conclusions?

-Are there concerns about ethical or regulatory requirements being met?

Reviewer #3: (No Response)

**Results**

-Does the analysis presented match the analysis plan?

-Are the results clearly and completely presented?

-Are the figures (Tables, Images) of sufficient quality for clarity?

Reviewer #3: (No Response)

**Conclusions**

-Are the conclusions supported by the data presented?

-Are the limitations of analysis clearly described?

-Do the authors discuss how these data can be helpful to advance our understanding of the topic under study?

-Is public health relevance addressed?

Reviewer #3: (No Response)

**Editorial and Data Presentation Modifications?**

Reviewer #3: (No Response)

**Summary and General Comments**

Reviewer #3: The authors have adequately addressed my earrlier comments. I have no further comments.

PLOS authors have the option to publish the peer review history of their article (what does this mean?). If published, this will include your full peer review and any attached files.

Reviewer #3: **Yes: **Collins Okoyo

---

## [Editor Report · Acceptance letter]

12 Apr 2024

Dear Dr. Wichit,

We are delighted to inform you that your manuscript, "Global seroprevalence of Zika virus in asymptomatic individuals: A systematic review," has been formally accepted for publication in PLOS Neglected Tropical Diseases.

Best regards,

Shaden Kamhawi

co-Editor-in-Chief

Paul Brindley

co-Editor-in-Chief
